# Efficacy of the reGENER@r Program on Socio-Emotional Skills and Sexist Beliefs in Perpetrators of Gender-Based Violence: A Pilot Study in Spain

**DOI:** 10.3390/bs14121194

**Published:** 2024-12-13

**Authors:** Ana Isabel Sánchez, Laura Maroto, Sara Rubiano, Clotilde Berzosa Sáez, Raúl Quevedo-Blasco, Karla Astudillo-Reyes, María Pilar Martínez

**Affiliations:** 1Mind, Brain and Behavior Research Center (CIMCYC), University of Granada, Campus Universitario de Cartuja, 18011 Granada, Spain; rquevedo@ugr.es (R.Q.-B.); mnarvaez@ugr.es (M.P.M.); 2Faculty of Psychology, University of Granada, Campus Universitario de Cartuja, 18011 Granada, Spain; lauramaroto@correo.ugr.es (L.M.); sararubiano@correo.ugr.es (S.R.); cberzosas@ugr.es (C.B.S.); karlastudillo@correo.ugr.es (K.A.-R.); 3Social Insertion Centre Matilde Cantos Fernández (Granada), General Secretary of Penitentiary Institutions, 18013 Granada, Spain

**Keywords:** gender-based violence perpetrators, reGENER@r program, gender role conflict, sexist beliefs, social-emotional skills

## Abstract

Background: Gender-based violence (GBV) is one of the most pronounced expressions of the unequal power relations between women and men. As a tool for action against this phenomenon, psychological intervention programs for perpetrators of GVB are offered. This is how reGENER@r was born; it is a two-month program based on psychoeducational and cognitive-behavioral strategies that is part of the alternative measures to GBV-related prison sentences. The purpose of this study is to assess the efficacy of the reGENER@r program on the variables of emotional intelligence, empathy, coping responses, emotional dependency, gender role conflict, and sexist beliefs. Method: To this end, a sample of 37 subjects convicted of crimes of GBV was collected, and a pre- and post-evaluation by means of self-report was carried out. Changes were examined for statistical significance and clinical significance. Results: Significant improvements were observed in the variables of cognitive avoidance, emotional attention, hostile sexism, and distorted thoughts about women and the use of violence. Conclusions: Limitations and implications of these findings are discussed, and some modifications are suggested such as making interventions longer, with a greater gender focus, adapted to the individual characteristics of the participants, and complemented with individual sessions.

## 1. Introduction

UN Women estimates that 736 million women (almost 1 in 3) have experienced physical or sexual violence by an intimate partner or outside the intimate sphere [1]. These figures highlight the extent of the problem of violence against women. Therefore, the United Nations General Assembly Resolution 48/104 [2] determined that countries should strive to eradicate this type of violence by promoting prevention, research, and the introduction of legislation to establish punishments against such violence. Meanwhile, Spain passed Organic Law 1/2004 on Comprehensive Protection Measures against Gender-Based Violence, which aims to eradicate discrimination against women through interdisciplinary work [3]. In this law, gender-based violence (GBV) is defined as an act of violence based on gender inequality that is committed by a man against a woman who is or has been his partner and includes any act of violence that occurs in a public or private setting [3]. Sexual, psychological, economic, physical, cyber, and vicarious violence all constitute types of GBV [4,5,6,7,8].

Despite the acknowledgment of GBV as a social and structural problem and as a public health issue [9,10], the efforts of governments and society to eradicate this phenomenon are far from being completely effective. Despite being defined in the law as GBV, Spanish jurisprudence tends to address this phenomenon more along the lines of IPV (intimate partner violence), but as the manuscript is based on Spanish law, GBV is still used as the preferred term.

That is why it is also important to explore the myths and attribution of responsibility in society for this and other offenses (such as sexual violence, see [11]).

In Spain, GBV continues to be a public health issue, an educational issue, and, above all, a structural issue. Regarding the number of women murdered in Spain, 32 have been recorded so far in 2024, with 58 murdered in 2023, totaling 1276 victims from 2003 to 5 August 2024 [12]. It is therefore important to determine what victims of GBV need in order to feel a sense of justice [13].

The number of men reported for GBV also increased by 9.7%, up to 36,434. Of these, 48.4% were between 30 and 44 years old. In 2023, 36,582 female victims of GBV were registered, corresponding to matters in which precautionary measures or protection orders had been issued. Almost half (47.8%) of the victims were between 30 and 44 years old [14].

Article 80 of the Spanish Penal Code provides for the possibility of suspending a prison sentence if it does not exceed two years in order to maximize the rehabilitative purpose that characterizes the Spanish justice system. This suspension is always conditional upon different requirements; the continuous and attentive attendance to a rehabilitation program for perpetrators of gender-based violence is an essential element for the suspension of a sentence in the case of GBV offenses. To develop adequate intervention programs, it is necessary to understand the individual characteristics of perpetrators of gender-based violence.

Allen et al. reported the importance of studying women’s violence, not only in the context of men’s violence but also within a broader sociocultural context [15]. Sexism is conceptualized by authors such as Glick and Fiske [16], who differentiate between hostile sexism (the most traditional and evident sexist belief that positions women as an adversary who needs to be dominated) and benevolent sexism (paternalistic attitudes based on the view of women as inferior beings who need to be protected) [17,18]. Distorted thoughts can lead to sexist beliefs that position women as inferior beings against whom it is justifiable to exercise violence [19,20]. Ferrer-Perez et al. [21] point out the importance of identifying irrational beliefs, attitudes, and distorted thoughts, not only among aggressors but also among the general population. Authors such as Herrero et al. [22], Juarros-Basterretxea et al. [23], and Guerrero-Molina et al. [24] have highlighted the direct relationship between GBV behavior, sexist beliefs, and justification of violence, while there is an indirect relationship between hostile sexism (a sub-variable of sexist beliefs) and the occurrence of GBV behavior.

On the other hand, abusers tend to adhere to their traditional gender role, which produces gender role stress if the situation demands otherwise. This gender role stress is directly related to the acceptance of GBV [25]. This construct includes four components that explore male socialization, emotional constraint, constrained affect among men, and work–family conflict [26]. This study was focused on perpetrators who were in a romantic relationship at the time, providing an example not only of GBV but also of IPV (intimate partner violence) IPV can be exercised by a man or a woman while in a romantic relationship.

Emotional variables have been of interest in understanding the genesis of GBV. Nyline et al. [27] state that, compared with a normative group, GBV offenders showed difficulties in identifying fear and sadness in other people. They even presented alexithymia, which implies that they would have difficulties in the identification, communication, and regulation of their own and other people’s emotions and would use aggressive behavior as a response [28]. Echeburúa et al. [29] identified differences between a group of GBV offenders and a normative group, where the first showed higher emotional dependence toward their partner, which correlated with depressive symptomatology. López and Moral [30] found a connection between emotional dependence toward the partner and aggression in participants attending a program of alternative measures.

Social skills, such as empathy and coping skills, are impaired in GBV offenders. Those offenders who have difficulty with emotional perspective-taking exhibit higher scores in psychological aggression, and those who show higher empathic stress (ability to share another person’s negative emotions) exhibit higher scores in physical aggression [31]. A recent study comparing aggressors with the general population found that intimate partner aggressors presented a different brain activation pattern in the presence of emotional stimuli, and this was associated with lower empathy and emotional regulation [32]. In relation to coping skills, few studies have been developed, although it is reported that—in relation to attachment styles—inadequate coping strategies can increase the occurrence of GBV [33,34].

Article 42.1 of Law 1/2004 states that the Penitentiary Administration will carry out specific programs for inmates convicted of GBV-related crimes. Regarding the efficacy of intervention programs for this type of offender, the literature shows partially contradictory data [35] between programs based on different models and participants who were court-ordered or voluntarily treated. In their meta-analysis, Cheng et al. [36] found that the programs led to positive (47.1%) and mixed (11.8%) changes in their studies, half of which had a quasi-experimental design and were based on the CBT or Duluth model. McNeeley [37] (with her quasi-experimental study with prison-sentenced participants) and Wilson et al. [38] (with their psychoeducational, quasi-experimental study with court-mandated perpetrators) did not find any changes between treated GBV offenders and control groups (McNeeley states that there were no changes between the effectiveness of the intervention inside prison or in the community). Despite the discrepancies in the literature, several authors agree on one point: intervention programs with GBV offenders are effective but the effect size is small [39]. As for the model on which to base programs, in their respective meta-analyses Fernández-Fernández et al. [40], Travers et al. [41], and Arce et al. [39] point out that cognitive-behavioral therapies exhibit the best results.

In terms of the recommended length of the interventions, Arce et al. [39] highlight the ineffectiveness of short interventions and even their counterproductive effect in comparison with long-term interventions. Moreover, these intervention programs are faced with several problems, particularly the cultural diversity of the participants, the presence of psychopathological disorders and/or substance abuse problems [35], and the low degree of motivation to participate in these kinds of programs [42].

In Spain, programs focused on GBV aggressors show positive results. It has been shown that the Intervention Program for Aggressors (PRIA) reduces sexist attitudes, impulsivity, and anger, increasing self-esteem, as well as emotional regulation [43]. The “Contexto” program is effective in modifying inmates’ conceptualization of violence against women, reducing their depressive symptoms, and increasing their participation in the community [44]. Finally, the “Galicia” program has been useful in increasing emotional recognition and regulation, modifying cognitive skills, and endorsing appropriate coping strategies [45].

Alternative measures programs have also been shown to be effective in the acceptance of liability, perceived seriousness of the offense, and decreasing the risk of reoffending [46]. However, there are no studies in Spain on the effectiveness of programs for sentences of less than 60 days. In this context, it is worth highlighting the reGENER@r workshop, aimed at men convicted of GBV offenses and sentenced to both community service (CS) and a maximum of one year of imprisonment. This workshop, based on the aforementioned PRIA program, was developed as a psychoeducational activity, in response to measure S241 of the State Pact against GBV of September 2017, updated and ratified in 2019 [47]. The reGENER@r workshop is centered on the risk–need–responsibility model (RNR) [48] and the Good Living Model [49], through a cognitive-behavioral and gender approach. Based on these approaches, the idea is that, just as violent behavior is learned, violent men can be taught alternative forms of adapted behavior in relationships. ReGENER@r works with CBT and psychoeducation, trying to eradicate sexist violence by going to the root of it—stereotypes, emotional issues, the culture of violence, and sexist beliefs, among others. ReGENER@r takes place in the community, with court-mandated participants, which allows them to put into practice everything that is taught in the workshops, in their day-to-day lives. This circumstance avoids the isolation and marginalization that prison can cause. Furthermore, techniques such as reinforcement of good behavior, giving homework, calling when a participant is absent, etc., are used in the workshop, which increases the participants’ motivation related to interventions.

The workshop is focused on the penal principle of reinsertion, which can be reflected in its intensity and brief duration, re-educating offenders convicted of GBV without the detriments of deprivation of liberty and from a new gender-based feminist approach that promotes equality [50]. In this study, we present the initial results regarding the effectiveness of the workshop on the variables previously proposed.

Thus, and considering the literature reviewed, the objective of the present study was to determine, among males convicted of GBV crimes, the efficacy of the reGENER@r workshop on the modification of emotional intelligence, emotional dependence, empathy, coping skills, the degree of gender role conflict, sexist beliefs, and justification of violence.

## 2. Materials and Methods

### 2.1. Participants

Participants were selected by the Social Insertion Centre [removed for blind review] personnel as part of the Services for the Management of Sentences and Alternative Measures (SGPMA) (General Secretariat of Penitentiary Institutions). This step had to be taken due to the nature of the alternative measure, which established mandatory attendance and participation for the subjects in order to avoid a prison sentence.

The eligibility criteria were: (1) males over 18 years of age; (2) current sentence for one or more GBV offenses; (3) sentence of up to 60 days of community service; and (4) attendance to the workshop and adherence to the rules of participation. Forty-one potential candidates who met the eligibility criteria were offered the possibility to participate in the study, of whom all but one agreed to take part. However, during the sessions, three of the subjects stopped attending or showed disruptive behavior during the course (non-participation in workshop tasks, as well as interrupting or sabotaging sessions) and were, therefore, excluded from the workshop and from the study. Participation and continuity in the workshop imply having the ability to respect the established rules and follow the instructions of each professional for the proper development and operation of the workshop.

The sample ultimately consisted of 37 participants who completed the workshop. It should be noted that two participants did not fully complete part of the evaluation protocol and were therefore excluded from the corresponding data analysis.

The final sample included 37 participants, all of whom were adult males, with a mean age of 40.91 years (SD = 13.34) (range 19–65 years old). Regarding marital status, 35% were single, 27% were married, 19% lived with a partner, and 19% were separated/divorced. In total, 81.1% of the participants were of Spanish nationality while 18.9% were foreigners. In relation to education level, 54% had completed secondary education, 16% had completed primary school, 13.5% had vocational training, 8% had no studies, 5.5% had university studies, and 3% had postgraduate studies.

As for the offenses for which they had been convicted, 64.9% were for the offense of battery defined in Article 153.1 of the Spanish Penal Code, 21.6% for the offense of threats defined in Article 171.4 of the Spanish Penal Code, 5.4% for the offense of coercion defined in Article 172.2 of the Spanish Penal Code, 5.4% for the offense of breaking a previous GBV sentence defined in Article 468.2 of the Spanish Penal Code, and finally, 2.7% for both threats and battery. The participants had received a mean custodial sentence of 36.89 days (SD = 12.13), the lowest custodial sentence being 10 days and the longest custodial sentence being 60 days.

### 2.2. Instruments

We used the Spanish version of the Trait Meta-Mood Scale-24, TMMS-24 [51,52]. The TMMS-24 is composed of 24 items that assess emotional intelligence through three components: emotional attention, clarity of feelings, and emotional repair from 1 (strongly disagree) to 5 (strongly agree), with a higher score indicating higher emotional intelligence. The Spanish adaptation has good internal consistency (Cronbach’s alpha was 0.90 for the attention and clarity components, and 0.86 for the repair component) [52].

The Partner Emotional Dependence Scale (SED) [53] is a 22-item questionnaire with a Likert-type response format ranging from 0 to 4, which assesses both current and past emotional dependence in partner relationships of at least 6 months. There is a direct relationship between scores and emotional dependence. This instrument presents adequate internal consistency (Cronbach’s alpha = 0.90) as well as high convergent validity [53].

The Cognitive and Affective Empathy Test (TECA) [54] has 33 items with a Likert-type scale ranging from 1 (strongly disagree) to 5 (strongly agree) representing four subscales. Two of these subscales evaluate cognitive empathy: perspective-taking (e.g., “I try to understand my friends by looking at situations from their perspective”) and emotional understanding (e.g., “I notice when someone tries to hide their true feelings”). The other two subscales evaluate both positive and negative affective empathy: empathy joy (e.g., “When something good happens to someone, I feel happy”) and empathy stress (e.g., “I cannot help but cry at the testimonials of people I don’t know”). The higher the score, the greater the empathy. This test has good internal consistency with a Cronbach’s alpha of 0.86 and a reliability coefficient by the method of two halves also of 0.86 [54].

We used the Spanish version of the Coping Responses Inventory-Adult Form (CRI-A) [55,56]. The CRI-A has 48 items and evaluates a person’s coping responses to problems and stressful situations with a Likert-type scale, ranging from 0 (“no”) to 3 (“almost always”). It is composed of eight scales measuring logical analysis, positive reappraisal, seeking guidance and support, problem-solving, cognitive avoidance, acceptance, seeking alternative rewards, and emotional discharge. Cronbach’s alpha for the Spanish version ranged from 0.50 to 0.70 [56].

We used the Spanish version of the Ambivalent Sexism Inventory (ASI) [16,57]. This inventory consists of 22 items with a Likert-type scale ranging from 0 (“strongly disagree”) to 5 (“strongly agree”). There is a direct relationship between the scores and sexist beliefs. This instrument is divided into two subscales: hostile sexism and benevolent sexism. The ASI has excellent internal consistency with a Cronbach’s alpha of 0.90 and 0.88 [57].

The Inventory of Distorted Thoughts about Women and the Use of Violence-Revised (IPDMUV-R) [20] has 21 dichotomous response items (“true/false”). It identifies thoughts or cognitive biases related to the inferiority of women or the legitimization of the use of violence against them. It has a good internal consistency (Cronbach’s alpha = 0.74) and establishes a cut-off point of 8 between potential aggressors and non-potential aggressors, establishing a direct relationship between distorted thoughts and the score obtained [20].

We used the Spanish version of the Gender Role Conflict Scale-Short Form (GRCS-SF) [26,58]. It consists of 16 items with a Likert-type response format ranging from 1 (“strongly disagree”) to 6 (“strongly agree”). Using four subscales (emotional restriction; restricted affections between men; success–power–competition; and conflict between work and family relationships), this instrument examines the discomfort caused by the incompatibility of the traditional male role assumed and the situational demands that require a break of this role. The higher the score, the more pronounced the conflict present. The Spanish version of the GRCS-SF has a Cronbach’s alpha of 0.75 [58].

We used the Spanish version of the Social Desirability Scale (SDS) [59,60]. This scale consists of 33 dichotomous response items (“true/false”) that measure how a person alters their image according to what is socially favorable for them. The higher the score, the greater the social desirability. It is considered a unidimensional scale and has a robust internal consistency in its Spanish version, with a Cronbach’s alpha of 0.78 [60].

### 2.3. Procedure

The reGENER@r workshop was developed in conjunction between the Autonomous University of Madrid and the General Secretariat of Penitentiary Institutions and consists of 10 topics to be addressed [50]. In the present study, the workshop was adapted to 8 sessions (32 h in total) to be in the range of minimum hours corresponding to the sentences of the participants. The workshop was applied in a group format for 2 months, with a weekly frequency (4 h per week). Four working groups were formed, initially composed of between 6 and 13 participants, and each group was taught by two therapists trained in gender issues and with experience in teaching intervention programs within prison communities.

Participants were informed about the research and filled out an informed consent form with information about the study. The present investigation was approved by the Ethics Committee of the University of Granada (see Institutional Review Board Statement).

The workshop addressed psychoeducational and cognitive-behavioral strategies. Table 1 specifies the content of each session, which was developed through theoretical explanations and group dynamics, with different homework assignments during the week. The fidelity of the workshop implementation was verified through a manualized intervention protocol [50] and periodic supervision carried out by the CIS program coordinator.

Prior to the application of the tests and the beginning of the workshop, the research objective was explained to every participant, and they were provided an informed consent form containing information about the study, the tests that would be administered, and the processing of the data, emphasizing the confidential nature of the personal information that would be obtained from them. The test application format was self-administered, with explanations by the therapists in charge, and with a paper record of the participants’ responses.

### 2.4. Data Analysis

Data were computed with JASP software (version 0.17.2.1) [61]. Basic descriptive statistics were expressed as mean (M), standard deviation (SD), and percentage (%). *p* < 0.05 was considered as a value indicative of statistical significance and 95% confidence intervals were applied. The Shapiro–Wilk test was used to examine the distribution normality of the data.

Changes at the statistical level were analyzed with Student’s *t*-test for related samples for measurements with normal distribution, and the Wilcoxon test was used for those with distribution deviating from normality. For Student’s *t*-test, the effect size was calculated with Cohen’s d, considering that values between 0.2 and 0.4 indicate a small effect, between 0.5 and 0.7 a medium effect, and above 0.8 a large effect [62]. For the Wilcoxon test, the effect size was computed with the paired rank biserial correlation (rB). Correlations were interpreted as follows: low (0.10 to 0.29), medium (0.30 to 0.49), or high (0.50 or higher) [63].

Following Lambert and Ogles’ [62] recommendations for outcome research in psychotherapy, the estimation of the clinical significance of changes was based on the Jacobson–Truax Method (Reliable Change Index, RCI) [64]. Participants were classified according to the categories of “improved” (RCI > 1.96 in the functional direction), “deteriorated” or “no change”. The RCI was calculated using the Leeds Reliable Change Indicator [65].

## 3. Results

The sample size (*n* = 37) of the study was adequate since the minimum required sample is *n* = 34 considering an *α* of 5% as the threshold probability to reject the null hypothesis being true (type I error), a *β* of 20% as the probability to accept the null hypothesis being false (type II error), an anticipated effect size of 0.50, and 1 SD of expected change in the outcome.

The working groups were matched on demographic and prison variables: age, *F*(3, 33) = 0.87, *p* = 0.468, educational level, *Χ*^2^(21) = 25.42, *p* = 0.229, marital status, *Χ*^2^(12) = 12.69, *p* = 0.392, nationality, *Χ*^2^(21) = 19.80, *p* = 0.534, and offense type, *Χ*^2^(18) = 18.45, *p* = 0.427.

The Shapiro–Wilk test showed that all variables had a normal distribution (values between *SW* = 0.95, *p* = 0.091 and *SW* = 0.98, *p* = 0.91), except for the overall TECA index; the CRI-A for positive reappraisal, cognitive avoidance, and acceptance or resignation; and the variables of the Gender Role Conflict Scale-Short Form (GRCS-SF), which showed a distribution far from normality (values between *SW* = 0.86, *p* < 0.001 and *SW* = 0.094, *p* = 0.045).

As seen in Table 2, there were significant changes between pre- and post-treatment in the variables regarding distorted thoughts about women and the use of violence (IPDMUV-R), hostile sexism (ASI), emotional attention (TMMS-24), and changes close to statistical significance in cognitive avoidance (CRI-A). The effect size for distorted thoughts about women and the use of violence (IPDMUV-R) was large, for hostile sexism (ASI) it was medium, for emotional attention (TMMS-24) medium, and for cognitive avoidance (CRI-A) medium. In the remaining measures, no significant changes were observed between pre- and post-treatment.

To determine whether significant variations were achieved at the clinical level, we used the RCI [64], which reports the magnitude of change in each participant. In the attention component of the TMMS-24, it was observed that, of the 37 participants, 8 achieved significant improvement, 27 showed no change and 2 worsened (see Figure 1). Regarding cognitive avoidance assessed through the CRI-A, it was found that 12 of the participants had improved, 20 remained unchanged and 5 showed deterioration (see Figure 2). For the variable hostile sexism (ASI), it was observed that 8 participants had significantly improved while only one had deteriorated, the other participants remaining without significant changes (see Figure 3). For the variable distorted thoughts about women and the use of violence (IPDMUV-R), 4 participants were identified with significant improvements while the remaining participants showed no significant changes (see Figure 4).

Social desirability scale (SDS) did not exhibit significant changes and was found mostly below the different cut-off points proposed in previous literature [66]. Considering the lowest cut-off point (25), only 9 participants showed indices of social desirability in the pre-treatment, while, in the post-treatment, only 6 were reported as showing signs of strong social desirability. However, if the highest cut-off point (27) is taken into account, only 6 participants showed signs of social desirability before starting the intervention and 3 once the intervention was over. These data were not significantly altered by the intervention and were not large enough to bias the investigation.

## 4. Discussion

The aim of this study was to evaluate the efficacy of the reGENER@r workshop with offenders of GBV in relation to the variables gender role conflict, sexist beliefs, emotional intelligence, emotional dependence, empathy, and coping with problems. ReGENER@r, a relatively new intervention program, innovative due to its gender focus and length, aims, among other objectives, to teach learning strategies for maintaining healthy and egalitarian romantic relationships [50]. At a general level, rehabilitation or re-education programs for perpetrators of GBV (such as PRIA or Contexto) show encouraging results [39,44,67], although research continues regarding the effectiveness of interventions with perpetrators [40], seeking to regulate methodological issues [35].

The results showed that after the application of the program, the participants’ cognitive avoidance as a coping strategy in problematic situations decreased, this being the predominant style in GBV offenders [39]. It was observed that at the end of the program, the participants perceived the problem more realistically, in order to find an adequate solution. Furthermore, total emotional attention (emotional intelligence) increased significantly, indicating that participants developed a better ability to understand and pay attention to their feelings appropriately.

Findings also revealed that the program produced statistically significant changes in the variables of hostile sexism and distorted thoughts about women and the use of violence. On the one hand, hostile sexism and distorted thoughts about women and the use of violence were reduced, this being one of the main axes that sustain violence towards women as a strategy for coping with couple conflicts [20]. This therefore generated a development of attitudes of greater respect towards women, egalitarianism in the interaction with them, and less justification of the use of violence as a conflict solution. Regarding the rest of the variables analyzed, no additional change was detected.

The reGENER@r program works more intensively on the eradication of sexist beliefs and elimination of violence in its core sessions, once the barriers of justification and minimization of criminal behaviors have been addressed [42], which could justify the greater internalization of the program’s core content. Since most of the studies that address this topic measure their overall success through the recidivism of the participants [68], it would still be considered too early to be able to satisfactorily measure the overall effectiveness of reGENER@r, as less than a year has passed since the end of the workshops, and the average time before conducting a hypothetical post-evaluation is usually 6 months, 1 year, or 18 months [39]. However, Pérez et al. [43] noted that the PRIA program (with an overall long-term effectiveness of 93.2%) [69] also reduced sexist attitudes and hostility, among other achievements, as did reGENER@r, which produced improvements in hostile sexism in its participants.

Examining the group-level changes obtained with the workshop in the Emotional dependence on the partner (SED) variable, an initial M of 42 was obtained and an M of 41.97 at the end of the workshop. According to the SED scales, a score between 10 and 21 indicates moderate dependence, between 22 and 26 high dependence and over 37 extreme dependence [53]. Considering the above, the participants continued to present emotional dependence in accordance with studies, such as that of Echeburúa et al. [29]. Furthermore, this emotional dependence has been found to be linked to insecure attachment types [70], so it would be recommended to add sessions that promote a healthier and constructive attachment in the bonds with partners in treatment programs. In the TECA, the total score pre-treatment *M* was 115.59 and the post-treatment *M* was 114.51, both corresponding to a percentile of 70 [54], indicating that the participants present high cognitive and affective empathy.

In line with the social stereotype of the GBV offender, Arce et al. [39] indicate that perpetrators tend to resort to a greater extent to maladaptive problem resolution such as resignation, emotional discharge, and alternative activities, leading them to avoid coping with the situation. This explanation is not in line with the results of the present study, where it can be observed that, in general, the coping strategies of the participants are adequate.

Finally, it is essential, among other variables, to assess the risk of GBV [71], suicidal ideation, and associated factors [72,73], or even to analyze the attitudes of professionals in interventions in cases of GBV [74].

The present study has some limitations. With respect to the reGENER@r program, the most notable limitation is its length, being a brief intervention that takes place over a maximum period of 2 months. According to Arce et al. [39], this affects the scope of the results since cognitive restructuring (one of the components of the workshop) requires more time to show its effects. Another important limitation, at the methodological design level, is the absence of a comparison control group. Likewise, the study did not assess the stability of the changes in subsequent follow-ups after the end of the intervention. Finally, the compulsory nature of attendance and participation in the program as an alternative measure to prison should be highlighted, which limits the willingness to participate in a follow-up after the end of reGENER@r.

For future research, it would be necessary to further examine the identification of the parameters related to the procedure or the participants that condition the success of the program in order to encourage greater involvement in the intervention and its contents, and, therefore, achieve a greater reduction of the factors related to GBV.

As proposed modifications, it is understood that, although the reGENER@r workshop generated changes in the social, emotional, and cognitive skills of the participants, adjustments should be made to promote more intense, deeper, and longer-lasting changes. The absence of homogeneity in the characterization of GBV aggressors makes it necessary to implement individual sessions in the workshop [75], along with different variants according to individual particularities (addictions, cultural differences, mental disorders, etc.) [76], in line with the risk–need–responsivity (RNR) model of Andrews et al. [48]. To strengthen the effectiveness of these interventions at the group level, it is recommended to extend the time horizon of the intervention, increase the motivation of the participants, intersperse group therapies with individual sessions, and adapt the intervention according to the psychosocial profile of the person [77,78].

On the other hand, the importance of the social normalization of certain sexist patterns and beliefs that have to be abolished must be emphasized, not only with better knowledge and greater awareness of the phenomenon of GBV but also by including a gender perspective in these programs for aggressors, thus facilitating their reintegration into the community and the development of more constructive family/partner bonds.

## 5. Conclusions

The present study shows that although the intervention is on the right track, adjustments must be made in the reGENER@r workshop to improve the results even more. These efforts would make it possible to implement more effective treatments in order to reduce the recidivism of GBV offenders.

## Figures and Tables

**Figure 1 behavsci-14-01194-f001:**
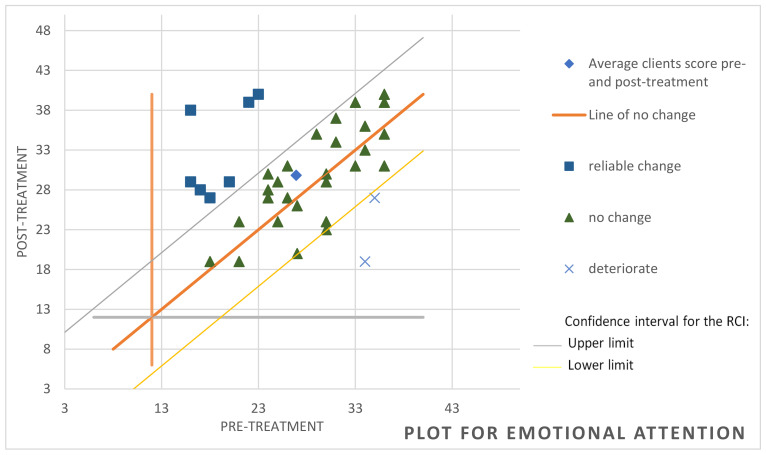
Individual changes in the emotional attention component in TMMS-24. Note. This figure shows the number of participants that improved, got worse, or remained unchanged after the workshop out of the total of 37. TMMS-24 = Trait Meta-mood Scale-24.

**Figure 2 behavsci-14-01194-f002:**
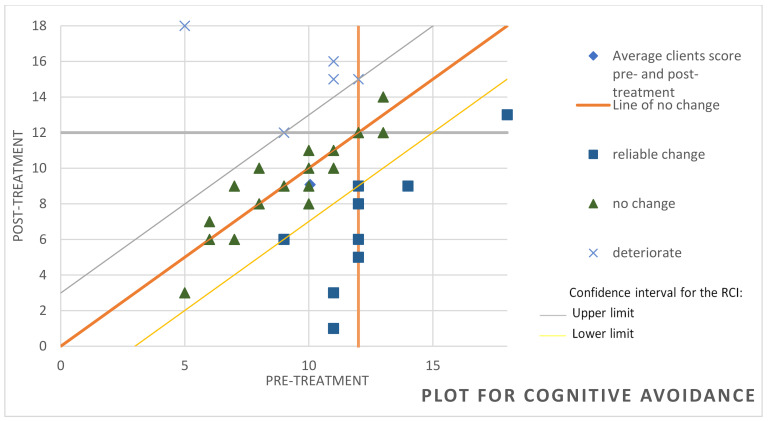
Individual changes in cognitive avoidance component in CRI-A. Note. This figure shows the number of participants that improved, got worse, or remained unchanged after the workshop out of the total of 37. CRI-A = Coping Responses Inventory–Adult Form.

**Figure 3 behavsci-14-01194-f003:**
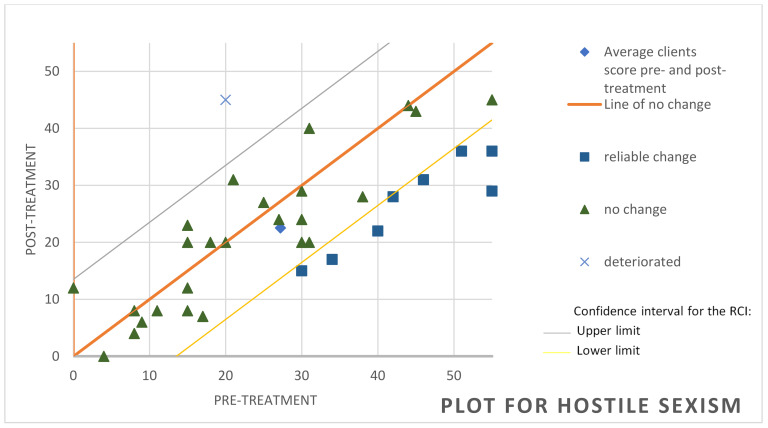
Individual changes in hostile sexism in ASI. Note. This figure shows the number of participants that improved, got worse, or remained unchanged after the workshop out of the total of 35. ASI = Ambivalent Sexism Inventory.

**Figure 4 behavsci-14-01194-f004:**
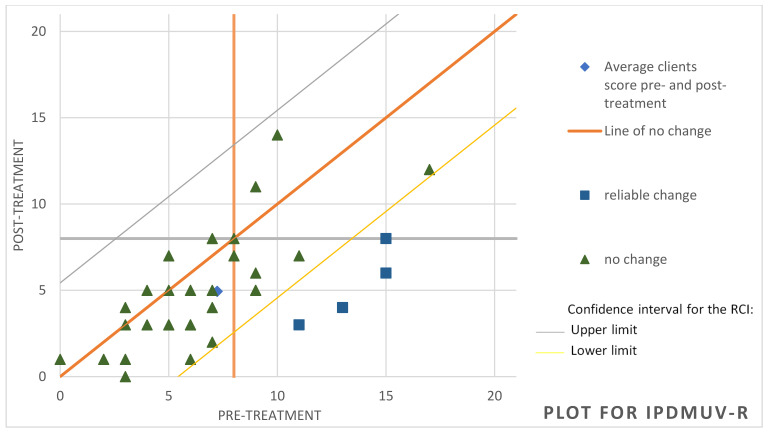
Individual changes in distorted thoughts about women and the use of violence. Note. This figure shows the number of participants that improved, got worse, or remained unchanged after the workshop out of the total of 35. IPDMUV-R = Inventory of Distorted Thoughts about Women and the Use of Violence-Revised.

**Table 1 behavsci-14-01194-t001:** Content of each session of the workshop.

Session	Variables	Content
1	Coping with emotions	Pre-treatment evaluation, presentation of norms and discovering our emotional world: identification and naming of emotions and description of emotional management strategies.
2	Gender stereotypes and adherence to the gender role	Exploring gender socialization and stereotypes and creation of new masculinities. Explanation of the diversity of gender role expressions and promotion of tolerance for atypical ones.
3	Adherence to the gender role and assumption of responsibility	Construction of new masculinities and breaking the spirals of violence. Promotion of tolerance among the group to make the diversity of gender role expression visible. Raising awareness of the consequences of gender-based violence.
4	Assumption of responsibility and justification of violence	Breaking the spirals of violence. Explanations about the different types of GBV and its consequences.
5	Sensitization, empathy, and jealousy	Encouragement of self-control strategies. Understanding jealousy, emotional dependence and its origins, and promotion of strategies to reduce them.
6	Emotional dependence, psychoeducation about healthy romantic relationships	Emotional dependence and healthy relationships. Analysis of behaviors that threaten autonomy and introspection regarding the participants’ relationships with their partners.
7	Consent, stereotypes, educational styles, assertiveness, and family responsibility	Positive sexuality and family co-responsibility. Explanation of myths about sexuality and family roles in relation to gender roles and raising awareness about them.
8	Couples and family quality relationships, recidivism prevention	Family co-responsibility, finding balance, and well-being. Reinforcement of what was learned in the workshops through dynamics. Post-treatment evaluation.

**Table 2 behavsci-14-01194-t002:** Descriptive statistics and changes in the examined measures.

Variables	Pre-Treatment *M* (*DT*)	Post-Treatment *M* (*DT*)	Student’s *t*/Wilcoxon’s *W*	*p*	Effect Size
TMMS-24 Attention	26.89 (6.41)	29.84 (6.14)	−2.38	0.023	−0.39
TMMS-24 Clarity	29.22 (6.72)	30.51 (6.51)	−1.12	0.270	−0.18
TMMS-24 Repair	31.11 (6.39)	31.59 (5.75)	297	1.000	−0.00
SED-Total	42.03 (19.88)	41.97 (21.40)	0.01	0.989	0.00
TECA-Total	115.59 (13.69)	114.51 (12.18)	378.5	0.302	0.20
TECA-Perspective taking	29.46 (4.95)	28.62 (5.32)	1.12	0.269	0.18
TECA-Emotional understanding	31.05 (5.11)	31.11 (4.22)	−0.06	0.950	−0.01
TECA-Empathic stress	22.89 (5.78)	23.16 (4.11)	−0.30	0.767	−0.05
TECA-Empathic joy	31.81 (5.42)	31.59 (4.65)	0.38	0.703	0.06
CRI-A-Logical analysis	10.27 (3.02)	9.76 (3.68)	0.58	0.568	0.09
CRI-A-Positive reappraisal	10.78 (2.93)	10.97 (3.98)	307.5	0.635	0.10
CRI-A-Seeking guidance and support	9.22 (3.23)	9.89 (4.15)	−1.02	0.313	−0.17
CRI-A-Problem solving	11.40 (3.14)	11.19 (3.58)	0.29	0.771	0.05
CRI-A-Cognitive avoidance	10.05 (2.70)	9.08 (3.71)	299	0.079	0.37
CRI-A-Acceptance	8.78 (3.38)	8.70 (4.11)	326.5	0.625	0.10
CRI-A-Seeking rewards	9.43 (3.36)	8.19 (4.83)	1.38	0.176	0.23
CRI-A-Emotional discharge	5.70 (3.89)	6.13 (3.64)	−0.61	0.547	−0.10
ASI-Total	48.46 (25.39)	44.83 (22.21)	1.27	0.212	0.22
ASI-Hostile sexism	27.20 (15.42)	22.54 (12.66)	2.69	0.011	0.45
ASI-Benevolent sexism	21.26 (12.39)	22.29 (11.26)	−0.64	0.524	−0.11
IPDMUV-R-Total	7.23 (3.84)	4.77 (3.01)	4.62	<0.001	0.79
GRCS-SF-Total	40.77 (12.05)	42.14 (13.54)	−0.65	0.524	−0.11
GRCS-SF-Emotional restriction	8.06 (4.14)	8.83 (4.74)	−0.98	0.336	−0.17
GRCS-SF-Success-power-competition	11.49 (3.86)	12.14 (4.53)	−0.96	0.347	−0.16
GRCS-SF-Conflicts between work and family relationships	12.34 (4.19)	11.66 (4.27)	0.78	0.440	0.13
GRCS-SF-Restricted affections between men	8.89 (4.43)	9.51 (5.56)	−0.79	0.437	−0.13
SDS-Total	19.37 (6.04)	19.34 (5.30)	0.04	0.969	0.01

Note. ASI = Ambivalent Sexism Inventory; IPDMUV-R = Inventory of Distorted Thoughts about Women and the Use of Violence-Revised; GRCS-SF = Gender Role Conflict Scale, Short-Form; SDS = Social Desirability Scale; TMMS-24 = Trait Meta-Mood Scale-24; CRI-A = Coping Responses Inventory-Adult Form; SED = Partner Emotional Dependence Scale; TECA = Cognitive and Affective Empathy Test.

## Data Availability

Data available on request.

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
