# Peer review of "Efficacy of the reGENER@r Program on Socio-Emotional Skills and Sexist Beliefs in Perpetrators of Gender-Based Violence: A Pilot Study in Spain"

_behavsci, 2024, doi:10.3390/bs14121194_

Round 1

Reviewer 1 Report

Comments and Suggestions for Authors

Thank you for your paper evaluating the efficacy of an intervention program targeting perpetrators of gender violence. Overall, I think there is some work to do regarding language, discussion of theoretical underpinning and explaining the results of the study but I can appreciate that valid contribution of such a program. 

General comments:

1. From what you said, it seems that Spanish law defines GBV as what most would characterize as IPV (i.e., current/past partner). I appreciate that definitions differ but because you use GBV extensively throughout your paper, it is necessary for you to be explicit that this is how you are defining GBV so that there is no confusion amongst non-Spanish readers.

2. There is no discussion of the theoretical underpinning of your work - how does theory contribute to the specific variables you've chosen (as opposed to others) and, secondly, to your expectation of behavioural change? All things being equal, why would this be successful where another program would not be?

3. Your discussion section needs more detail, you offer short explanations for the significant variables but nothing for those that were not significant. Is this because you should have chosen different variables or for some other reason (e.g., time in study, participant demographics etc.)?

Specific comments:

Title: 

1. gender violence? Would 'gender-based violence' be more appropriate?

Abstract:

1. "victims and complaints is still high" - I'm not sure about the blend of these two words in this context, firstly, and secondly, some might argue that it is not that more VAWG (or the same) is occurring, rather that increased public discourse means that those who did not report before now feel more comfortable/able/knowledgeable to report. I appreciate this is just the abstract but, as it sets the scene, it might be reconsidering this phrasing.

2. "batterers" - this seems an incongruous word choice. Firstly, not all violence is physical, secondly, this term seems rather colloquial in the context it is being used, and, thirdly, it might be more cohesive to be consistent in terminology (i.e., perpetrators of...) to ensure that your message remains clear.

Introduction:

1. I'm not sure I fully understand the first paragraph. The mention of myths and attribution doesn't quite fit with what you've said. Perhaps offer a bit more context to make this point clearer. Especially as the paragraph that follows doesn't really relate to attitudes/responsibility.

2. What do you mean by "precautionary measures issued"? I looked at the relevant reference and this makes a bit more sense but try to be clearer that you're referring to protection orders (a slightly more recognizable term) than something else.

3. "gender-based aggressors" or "gender aggressors" - I'm not sure this is appropriate terminology, it seems a bit awkward in phrasing.

4. "Historically, research in this area has largely omitted..." - I'm not sure this is entirely accurate. It is my understanding that such variables are often present in the literature. If I have misunderstood, then I would suggest you include a more updated reference, perhaps a meta analysis or systematic literature review, that supports this.

5. Though I appreciate you offering the distinction, based on what you included about Spanish law, it sounds like GBV is characterized as IPV because the perpetrator must be a former or current partner. For clarity's sake, it might make more sense for you to include this more explicitly.

6. It sounds like you're saying that the difference between your program and the other programs is, primarily, the lack of incapacitation but this is presented in a bit of a waffley manner. I would try to be more explicit about that and indicate why this is a preferred method than the current programs.

7. There seems to be no clear discussion of theoretical underpinnings (this would cover the point above) - why do you think your method is likely to be more successful than others? What does theory say about those variables and why/how they reduce GBV/support behavioural change?

Methods: 

1. You might want to explain some of the terms you use "empathy joy", "empathy stress", for example. This could be helped by including an example statement for the different instruments.

Findings:

1. "Regarding the rest of the variables..." - this doesn't really seem sufficient. Based on how you've phrased it, I found it a little difficult to follow what had been significant and what was not, but after going back to your methods, it seems like you had five significant changes (emotional intelligence, coping strategies, hostile sexism (not benevolent), distorted thoughts women and use of violence). That leaves at least five (six if you count benevolent sexism) variables that were not significant. Why is this not discussed? You go on to say about the internalization of the program's content but then why weren't further changes identified?

2. You mention that participants showed "extreme emotional dependence", is this why there wasn't a significant change or was it for some other reason? You mention about insecure attachment but this is something that could have been explored more deeply here and in your introduction.

3. You say that the participants possessed adequate coping strategies which is in conflict with previous research. Why might this be the case? What does research/theory say about this?

4. I'm not sure it's appropriate to direct readers to another paper without fully explaining role differences or stereotypes yourself.

5. "Finally, studies aimed at forensic psychological..." - what does this mean? At most, this seems like a limitation but it's not clear how it fits in with what you did/didn't do.

Limitations:

1. Why was this length of time chosen? You make it clear that research suggests it should be longer so it would be helpful to offer a bit more insight in your 'procedure'.

2. Why did you decide not to use a control group? It's fair to identify this but there's no explanation offered.

Future research:

1. I was surprised by the focus of this. Though you identified some significant changes, approximately the same number of variables were not significant. To then suggest that you should consider this program longitudinally without adding a bit of context (i.e., perhaps the program needed to be longer) seems a bit idealistic. I know you mentioned this in the limitations but it would appropriate to reiterate that here rather than focusing on maintenance of changes.

2. Some good sessions on modifications.

Conclusion:

1. Improve the quality of life of offenders? I'm not sure you've mentioned this, it seems an odd comment.

Comments on the Quality of English Language

Generally, the quality of English language was good but I did catch one Spanish word (de, line 169). There was some language and phrasing that was a bit awkward and could be improved but I wouldn't say there were major errors.

Reviewer 2 Report

Comments and Suggestions for Authors

The manuscript presented is well structured and shows a lot of potential, although it still needs a lot of work. The language used is quite clear. The topic addressed is extremely important, which is quite clear from the presentation of statistics and the framework provided in the introduction.

At the introduction level, the paragraph covering statistics related to men could be clarified. There is talk of the number of men reported, but it is not clear whether we are talking about male victims (which will not make much sense as we are talking about gender-based violence), male aggressors or another possible category. Also the conflict between traditional gender-role adherence and situational demands could be better explained, as it is not clear to the reader why this conflict could lead to GBV. Additionally, when addressing the effectiveness of intervention programs (in the paragraph starting in line 105), there is a lack of information on what are the common factors and distinctive factors of the intervention programs addressed in the studies cited - information that can be crucial to understand the differences in terms of study results. The same can be said when addressing intervention programs in Spain (paragraph starting in line 120).

The method chapter is clearly structured, allowing an understanding of what was carried out in the study. In terms of inclusion criteria and participant collection, I would like to understand what the authors mean by "disruptive behavior", as it can be an important highlight for understanding the composition of the sample. Furthermore, it is a little confusing to mention that the final sample was 37 participants only to mention that 2 people were excluded for not having completed the evaluation protocol. I also don't know if it wouldn't be of interest, when characterizing the sample, to mention how many participants were in the relationship in which the abuse occurred, so I leave this reflection to the authors' discretion. The mention of approval by the ethics committee should be included in the section of the manuscript relating to the procedure. In the instruments section, it is unclear whether the internal consistency values ​​presented are from the Spanish adaptation of the scales or whether they were calculated based on the sample in the present manuscript. If the values ​​are from the Spanish adaptations, the internal consistency values ​​calculated based on this sample are missing and must be added for each scale and subscale used in the data analysis. If the values ​​are from this manuscript, the value of 0.50 in the CRI-A is worrying, as it is an unacceptable value in terms of internal consistency. In the procedure section, I would like to see the program - which is the central focus of this manuscript - described in greater detail, not only at the level of content but also at the level of activities carried out.

In terms of results, these are extensively described, which is a positive point. Nevertheless, I believe I am missing Shapiro-Wilk test results for all normally distributed variables. When describing the results of comparison of means between paired samples, it will not be necessary to repeat the values ​​obtained by the tests in the text, as they are already present in table 2 and make it difficult to read the paragraph that starts on line 292. Furthermore, you will find information that already concerns the interpretation of the results obtained (for example, the paragraph that begins on line 307) is included throughout the section, so this same information must be present in the discussion chapter and not in the results chapter.

As for the discussion chapter, it lacks any reflection on potential reasons why there were no differences in terms of various variables used. Furthermore, paragraph 410 seems to be poorly contextualized (participants showed extreme emotional dependence, but this did not change during the program?), requiring some development or reformulation.

Throughout the manuscript, more specific issues could also be improved, such as avoiding paragraphs that contain only one sentence (for example, the paragraph starting on line 74 could be added to the previous paragraph). It seems to me that there is an error on line 169 ("of 18.9%"), on line 197 (one extra parentheses), on line 213 (the sentence that starts the paragraph is unfinished, on line 229 (I believe they meant "The higher the score"), in line 328 (the sentence referring to the score is written in a confusing way and appears unfinished).

As I mentioned initially, I believe that the manuscript presented does have potential, so I congratulate the authors for the work already done and leave a word of encouragement to reflect better on it and potential improvements to be made.

Round 2

Reviewer 1 Report

Comments and Suggestions for Authors

Though the authors have responded to some of my comments with changes to the text, many of the responses to my questions have been answered directly back to me. Usually, I would have expected a response that either highlighted how the authors planned to make the change or, one that made it clear why they thought the change was not necessary.

I completely understand if the authors don't agree with the suggestions or decide, for reasons, not to make them but a lack of any indication was surprising.

In some instances, it did seem like the authors had misunderstood my comments, based on their response. I have included those below and tried to rephrase where possible but I would say this applies to most of my original comments.

1. The authors' response to this comment didn't match my comment. My comment was pointing out that Spanish law appears to define GBV in a manner that is more closely aligned to IPV (rather than GBV) so it would be necessary for the authors to make this clear because they then continue to use GBV. I appreciate that they have defined GBV and I commend them for that, I was merely suggesting that they include a line that makes it clear that this particular definition is more closely aligned to traditional definitions of IPV (or something to that effect).

2. There is still no real mention of theory in the paper. The authors contend that gender violence can be reduced/eradicated by a multidimensional approach - they don't explain how or why this would work. What theoretical basis is there for this to be successful? Further, what is the theory that underpins those specific variables? There is some explanation (e.g., brain activation pattern) but this is not consistent. To be clear, I'm not saying it would or wouldn't work, I'm saying that this is not in the text.

3. I'm not sure how to re-clarify this.

As I say, I've included a few above but, based on the authors' responses, it seemed that quite a few of my comments were misinterpreted so it might be worth reading through them all. I also wanted to reiterate that my comments were posed as questions for the authors to consider adding to the text or, letting me know why they didn't think it was appropriate to add, rather than telling me the answer.

Reviewer 2 Report

Comments and Suggestions for Authors

The manuscript underwent important revisions and was substantially improved. However, important issues remain to be resolved.

When comments about potential clarifications are made (e.g., regarding traditional gender roles and situational demand, disruptive behavior, sample size), the objective is not to clarify them only for the reviewer but rather in the article, so some Things were missing and I therefore encourage the authors to revise once again.

The internal consistency values ​​calculated with the sample under study are missing.

Round 3

Reviewer 1 Report

Comments and Suggestions for Authors

Thank you for your response to my comments. I think my comments have been largely addressed.